# Effect of High Potency Growth Implants on Average Daily Gain of Grass-Fattened Steers

**DOI:** 10.3390/ani9090587

**Published:** 2019-08-21

**Authors:** Rodrigo Arias, Cristobal Santa-Cruz, Alejandro Velásquez

**Affiliations:** 1Instituto de Producción Animal, Universidad Austral de Chile, Valdivia-Chile, Valdivia 5090000, Chile; 2Centro de Investigación de Suelos Volcánicos, Universidad Austral de Chile, Valdivia 5090000, Chile; 3Private consultors, Frutillar 5620000, Chile; 4Escuela de Agronomía, Universidad Católica de Temuco, Temuco 4780000, Chile

**Keywords:** beef cattle, grazing, anabolic

## Abstract

**Simple Summary:**

Improving efficiency in beef cattle production requires the adoption of technologies that are low cost but high in return. High potency growth-promoting implants (HGPs) are widely used under feedlot conditions but there are few reports of their use under grazing conditions. We conducted a trial to assess whether the use of high potency HGPs have advantages for grass-finished cattle. Our results were similar to those reported for feedlot cattle, showing that the quality of the pastures was good enough to take advantage of the HGP technology, generating a marginal income that justifies the use of technology under grazing conditions.

**Abstract:**

High potency growth promoter implants (HGPs) are widely used under feedlot conditions but there are few reports under grazing conditions. The study’s goal was to assess the effect of HGPs on the average daily gain of steers fattened in pastures. A total of 57 crossbreed steers (Hereford × Angus)—initial body weight = 356.65 kg ± 5.04 (SEM)—were randomly allocated to one of three groups: Control without HGP (n = 19), Synovex group (n = 17), and Revalor group (n = 21). The fattening period was 67 days using paddocks of *Lolium perenne* L. and *Trifolium repens* L. Body weight was recorded three times in the period. The data were analyzed using an analysis of covariance with a level of significance of 5%. The average daily gain (ADG) (1.55 ± 0.07 and 1.48 ± 0.09 kg/d) and the total weight gain (103.4 ± 4.9 and 99.2 ± 5.8 kg) were similar for Revalor and Synovex, respectively (*p* > 0.05). Moreover, HGP groups showed higher ADG and total weight gain (*p* < 0.01) than the control group (ADG = 0.93 ± 0.08 kg/d and a total weight gain of 62.2 ± 5.2 kg). Final body weights were 527.8 ± 8.5 kg and 512.2 ± 9.9 kg for Revalor and Synovex, respectively; and 479.9 ± 10.1 kg for Control. In conclusion, grazing-finished steers showed better performance when high potency HGPs were used, improving the ADG and final live body weight, generating a marginal income that justifies the use of the technology under grazing conditions

## 1. Introduction

The use of anabolic growth promoter implants (HGPs) in feedlots is a production technology that is considered routine in many parts of the world. Its use is based on the improvement in growth rate and feed conversion, as well as because it reduces the costs of live weight gain [1]. HGPs improve daily weight gain (ADG) by 15 to 25% and feed efficiency by 10 to 15% when compared to non-implanted cattle [2]. It also increases feed intake (6%), carcass weight (5%) and ribeye area [3]. Consequently, animals have a heavier carcass weight, while the benefits associated with the efficiency of feeding make a systematic and regular implant program profitable for the producer. In the United States, the use of HGPs increase income in the range of US $50 to $210 per animal, depending on the strategy of implants used, that is, the number of implants used and its potency [3]. In addition, there is a negative public perception of beef production from an environmental standpoint, particularly due to the contribution of the sector to climate change [4,5,6].

In Chile, we estimated that this technology could generate returns of US $58 to $100 per animal, considering a single implant in the fattening phase. The difference in the returns is in part due to the USA payment system, which considers beef quality, while in Chile the main drivers are live weight and age. Although this technology has numerous advantages, it has also been reported to have negative effects on marbling and meat tenderness, compromising quality [3].

Most of the research regarding HGPs has been carried out in intensive systems with high energy density diets (grains or by-products of grains), with few studies assessing HGPs with diets rich in roughages. In most of these studies, the roughage source used is silage [7,8]. Other studies have evaluated the HGPs with grazing cattle but in the stocking stage, previous to the fattening but usually with HGPs of moderate potency [9]. In summary, there are few studies assessing the effect of HGPs of high potency, on steers performance during the fattening stage but under grazing conditions [10,11]. It has been pointed out that under grazing conditions improvement in the ADG is achieved ranging from 8% and 18% as a result of the differences in botanical composition and the quality of the pastures [12]. Because grasslands have a lower energy density compared to a feedlot diet, it is recommended to use HGPs of moderate potency, while those of high potency should be used in animals that consume high energy diets (rich in grains) [12]. Consequently, the general objective of the study was to compare the effect of two types of high potency HGPs on the average daily weight gain of steers fattened under grazing conditions and the respective marginal economic impact.

## 2. Materials and Methods 

This research was conducted on a commercial farm in Southern Chile. All the procedures, including humane animal care and handling procedures, followed national legislation (Law No. 20,380 on Protection of Animals; Decree No. 29 about regulations for the protection of animals during their industrial production, their commercialization and in other areas that hold animals), whose application is supervised by the National Service of Agriculture and Livestock (SAG), the competent authority in this matter.

### 2.1. Animals and Facilities

The study was conducted at a commercial farm located 8 km north of Frutillar city, in the same municipality. A total of 57 crossbred steers (Hereford × Angus) acquired in a private purchase were used. All steers were subjected to the same health program and handlings. Upon arrival (10 March 2018), steers were weighed (356.65 kg ± 5.04 SEM), dewormed (Levamisol 7.5% at 1 mL/12.5 kg live weight (LW)) and then kept in a separate group within the farm. In addition, they received an anti-clostridial vaccine, following the dose recommended by the manufacturer. One week later, steers were castrated by means of an emasculator, without anesthesia. The deworming program continued with Ivermectin (1 mL/50 kg LW). During this period (prior to being implanted), the animals’ diet was based on pastures dominated by ryegrass, orchard grass, fescue and sweetgrass and clover, plus harvested forage (haylage bales) manufactured at the same farm. On September 22nd, steers received the third deworming treatment (one dose of Ivomec-F at 1 mL/50 kg LW, plus Ivermectin 10 mg + Clorsulon 100 mg). In addition, they received the anti-clostridial vaccination recommended every 6 months. At the same time, steers were weighed and randomly assigned to one of the following treatments: Control = steers without hormonal growth-promoting (HGP; n = 19); Synovex = steers implanted with Synovex Plus^®^ containing 200 mg of trenbolone acetate and 28 mg of estradiol benzoate ((Zoetis, Chile); n = 17); and Revalor = steers implanted with Revalor^®^ containing 140 mg of trenbolone acetate and 20 mg of estradiol benzoate ((Intervet, Chile); n = 21). 

All weighing procedures were done early in the mornings, three times during the period started at 22 September 2019 and finished at 28 November 2018. Each steer was weighed individually by using the digital balance (Iconix model FX 1). Data were recorded in an Excel spreadsheet for further analysis. During the experimental period, cattle grazed pastures of ryegrass (*Lolium perenne* L.) plus white clover (*Trifolium repens* L.). The nutritional characteristics of these pastures are presented in Table 1. The feeding management consisted of daily grazing strips, with an estimated supply of 11 kg DM per steer.

### 2.2. Economic Marginal Analysis

The marginal economic analysis was carried out considering the differences in weight obtained between the implanted versus not implanted steers (final LW–initial LW). Because control of dry matter intake was not feasible, neither DMI nor Conversion was analyzed. Likewise, the cost of HGP plus the labor were estimated at US$ 3.2 per animal. The price per kilogram of LW was obtained from a local Auction (US $1.7 per kg of live BW) for the time when animals were sold on 26 November 2018 (http://www.fegosa.cl/preciososorno/precioosorno.html).

### 2.3. Data analysis

The experimental model used corresponded to a completely randomized design with three treatments. Each animal was considered an experimental and observational unit. The level of significance was 5% and the statistical model evaluated was: Y_ij_ = μ + α_i_ + ε_ij_; where μ corresponds to the mean; α_i_ represents the effect of i^th^ treatment and ε_ij_ the experimental error associated to the i^th^ animal of the j^th^ treatment. All data were analyzed with the statistical package JMP v14.0 (SAS Institute, Charlotte, NC, USA).

## 3. Results

Weather conditions during the period of study are presented in Figure 1. In general, ambient temperature (AT) was lower in September but increased over time. Contrary, rain precipitations were higher in September and decreased over time. Those weather conditions allowed an average rate growth of the pasture of 35, 40 and 60 kg DM/ha/d for September, October and November, respectively. However, the combination of lower AT and presence of moderate but continuous rain events during the first weeks of October resulted in a reduction of 24% the growth rate of the pasture when compared with the monthly average. 

The nutritional characteristics of the pastures (Table 1) were estimated from Anrique et al. [13]. The content of CP of the pasture duplicates the requirements for a fattening steer. According to the National Academies of Sciences and Medicine [14], CP requirement varies between 10 to 12%. During the spring season, the metabolizable energy (ME) and net energy (NE) contents of the pastures are slightly lower than those observed in small cereal grains. The observed values of NFD and AFD in addition to literature reports for the area indicate a high digestibility value of the organic matter, that is, a forage of very good quality and consequently with a high consumption potential.

A similar pattern of LW gain was observed for the three groups until the time of implantation (Figure 2). Although the LW gain was sustained over time, there was a lower average daily gain (ADG) for the period after HGP application (between 22 September 2018 and 10 May 2018), which was more noticeable for the control group. This coincides with the aforementioned adverse climatic conditions, characterized by rainfall and low ambient temperatures, which decreased the forage intake of the animals. The highest rates of ADG were observed from October ongoing, particularly between steers with and without HGPs, coinciding with better climatic conditions that favor both grass growth and forage intake.

Likewise, ADGs were not homogenous through the study (Figure 3), observing an interaction (*p* = 0.006) between days after implanting and type of HGP. In addition, no differences were observed between the two HGPs used on any of the days after-implanting (*p* > 0.05). At the end of the study period, all the steers with HGPs had a similar ADG (1.55 kg/d ± 0.07 SEM vs. 1.48 kg/d ± 0.09 SEM, for Revalor and Synovex respectively). Alike, the final body lives weights of steers that received HGPs were also similar (527.8 kg ± 8.5 SEM and 512.2 kg ± 9.9 SEM, for R and SP, respectively). Meanwhile, both groups had higher final body live weight than the control group (*p* = 0.002).

The marginal income per steer as a result of using HGP (average) was 60.1% higher than the control group (Figure 4), that is, US $63 more than the control group ($104–$167). Thus, this technology generates a return of approximately 19.7 times its value.

## 4. Discussion

The benefits of using HGPs in the fattening phase under feedlot conditions have been extensively documented in countries ass Australia and USA. The best performance is explained by the sensitivity of the muscles to the HGP. This response is differentiated, being greater in the extensor muscles associated with the long bones. Indeed, HGPs that combine estrogenic and androgenic hormones produce a greater response than implants of a single hormone [15,16]. It has been described that estrogen stimulates the production and release of hepatic somatotropin and also insulin-1-like growth factor (IGF1). Together, these secondary hormones stimulate the accumulation of muscle proteins. On the other hand, androgens act mainly through direct action on muscle tissue, stimulating protein synthesis and reducing muscle catabolism [17]. Likewise, HGPs that combine estradiol and trenbolone acetate result in a greater number of muscle satellite cells, a greater expression of IGF-1 mRNA in muscle tissue and an increase in circulating IGF-1 levels [18]. They can also affect the rate of proliferation and/or the activation status of satellite cells [1]. These cells provide the nuclei needed to support the hypertrophy of postnatal muscle fibers and are critical to determining the speed and degree of muscle growth [17]. It should be noted that the number and size of muscle fibers present in muscle tissue play an important role in determining the rate and efficiency of muscle growth and feed conversion [15]. Nevertheless, there is also reports of undesirable animal behavior associated with the utilization of implants that contains trenbolone acetate [19]. Likewise, there is evidence of negative effects on carcass quality, mainly related to marbling meanwhile, the results in tenderness have been variable [3].

Pasture fattening systems are characterized by exclusive grazing during the spring and/or the autumn, depending on the geographical area and weather conditions. Meanwhile, during the winter and summer period, temperature and humidity restrictions affect the growth and quality of the pasture. During these periods the animals continue grazing but they are usually supplemented (ad libitum) with harvested forages (silage and/or hay) and with or without small cereal grains, which are supplied in a ratio of 1.0% to 1.5% of animal LW. Normally, in Chile, beef breeds under these conditions reach ADG ranging between 0.8 to 1.5 kg/d for the spring season with final LW of 418 to 472 kg [20]. The observed ADGs herein were higher than those reported by Goic [21] for a barley silage rotation and silage system, that averaged in two seasons of 1.16 kg/d but there are in the above-mentioned range. Hojas [22] compared the effect of two HGPs with respect to the control without HGP in heifers of two genotypes (Hereford × Angus vs. Friesian) fattened in permanent pasture fertilized. The author stated the use of daily grazing strips through the use of electric fence by a 43 days period. The HGPs used had: (a) 200 mg of testosterone propionate + 20 mg of estradiol benzoate; and (b) 140 mg of trenbolone acetate and 36 mg of Zeranol (Component EH and Ralone^®^, respectively). The author concluded that the ADG of heifers with HGPs was higher than the control group (11.6%) but both were similar to each other. Based on this information we can argue that the quality of the pasture during the spring season is enough to respond to the use of high potency HGP. However, one of the main challenges is the adequate grazing management that allows avoiding sudden changes in the pasture quality that can be ascribable to inadequate management decisions. Thus, the interaction of all factors including weather conditions, animal, pasture and the proper management decision must be considered [23].

On the other hand, ADGs herein reported were higher than those reported for animals grazing ryegrass at the USA (database of TBA implants) and by Farney and Corrigan [24]. However, it should be noted that in the USA study, HGP of a lesser potency was used in comparison to those used in our study. In addition, the USA study assessed the backgrounding phase and not the fattening phase as in our case. For example, a recent study compared the effects of Revalor G (40 mg of trenbolone acetate and 8 mg of estradiol) and Synovex One Grass (150 mg of trenbolone acetate and 21 mg of estradiol benzoate), a new implant with a different coating technology, in stocker steers on a 90-d grazing season. No differences in ADG based on implant type (1.16 kg/d for Revalor-G and 1.13 kg/d for Synovex One Grass), were reported by the authors [24]. Berthiaume et al. [7], evaluated the effect of HGPs in animals reared and finished with a diet based on forage (pasture silage) but with no grazing. These authors concluded that ADG of fattening animals receiving HGP was 60% higher than those without HGP (1.18 vs. 0.74 ± 0.09 kg/d, respectively). This value was quite similar to the 60.1% reported in our study. In addition, the authors stated lower hot carcass weight (7.8%) and a lower quality (31%) for the animals that were not implanted. Thus, the authors conclude that in order to achieve the same economic return the non-implanted animals must be sell to price that is 16% greater than those implanted steers.

Regarding the slow LW gain observed in all treatments (Figure 1) between the implanting day and the first weight control, this could have been due to the combined effect of adverse weather conditions and the stress that may have resulted from going through the squeezing chute. In addition, those weeks were characterized by being unfavorable for the development and growth of the pasture, since November was colder than normal. Likewise, the air and soil temperatures were also lower than normal, which translated into a lower accumulation of day degrees with respect to a normal year.

To our knowledge, at present there are no reports of the effects of high potency HGP for cattle fattened under grazing conditions, since usually the implant potency is adjusted to the energy level of the diet, which is usually higher in feedlot diets than under grazing conditions. According to this argument, it is proposed that high-potency implants are destined for feedlot cattle, whereas the low and moderate potency implants are better for grazing cattle but unusual for the backgrounding stage. In the USA, technicians, often recommend moderate potency HGPs from late spring to mid-summer. In both mid-summer and fall, when pasture energy levels are low, they recommend low-potency implants. It should be noted that the potency of the implant is determined by both the type of activity (androgenic vs. estrogenic) and the concentration of the hormones. Consequently, practically any implant containing TBA (40 to 200 mg) is considered of high potency, while those based in estrogen (8 to 20 mg estradiol benzoate) are of lower potency. Finally, the utilization of this technology not only implies a better performance of the animal but also an improvement in the environmental and economic sustainability metrics [25], in a context of growing global beef demand, particularly from intensive grazing systems. Thus, zones with high-quality grasslands like southern Chile, Argentina and Uruguay have an important role in supplying beef. In fact, it has been estimated that intensive grazing systems supply about 20% of global beef production [6].

## 5. Conclusions

Under the conditions in which the study was developed, it is possible to conclude that steers under grazing conditions receiving a growth-promoting implant achieved a greater average daily weight gain than those that were not implanted. Likewise, there were no differences between comparative HGPs, and its utilization generates an important economic return to cattlemen.

## Figures and Tables

**Figure 1 animals-09-00587-f001:**
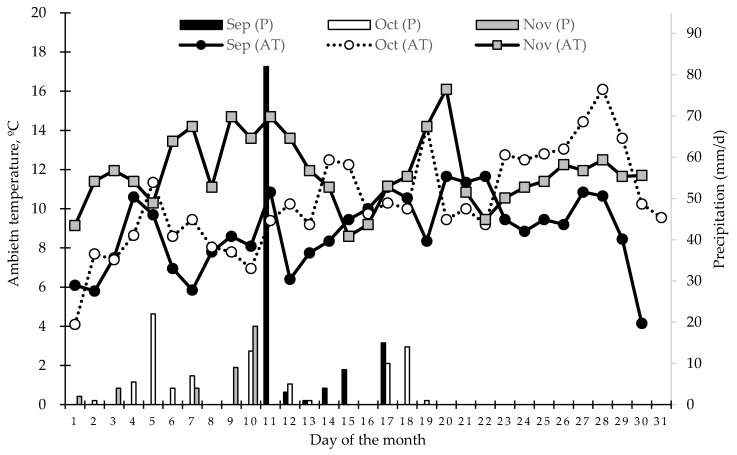
Records of ambient temperature (AT, lines, at the left axis) and daily rain precipitations (P, bars, at the right axis) for the study period. Data provided by the meteorological station of Cranberries Austral Chile S.A., located 3.5 km from the study location.

**Figure 2 animals-09-00587-f002:**
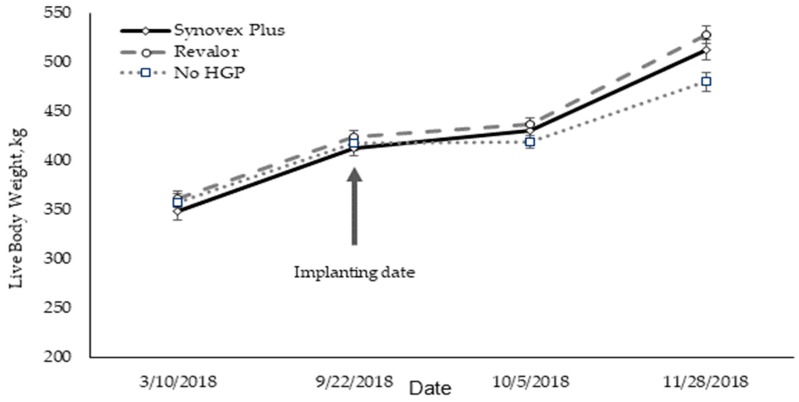
Evolution of body weight gain of steers during the study period (SEM) (HGP = hormonal growth promoting).

**Figure 3 animals-09-00587-f003:**
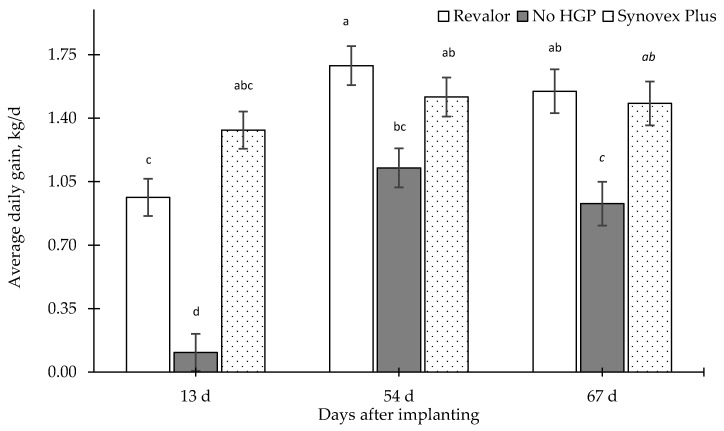
Average daily weight gains (kg/d) of steers according to the number of days after growth promoting implantation (SEM). Different letters indicate differences between groups according to Tukey’s test (*p* < 0.05) (HGP = hormonal growth promoting).

**Figure 4 animals-09-00587-f004:**
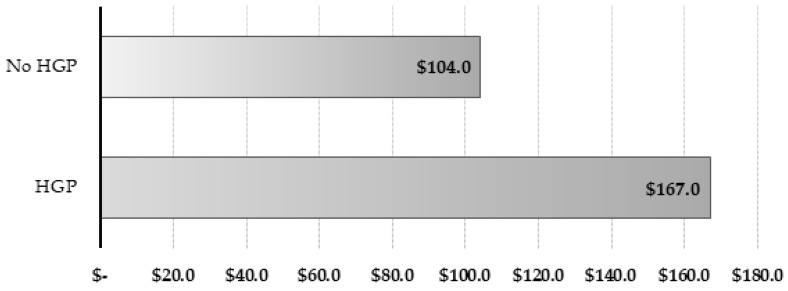
Marginal income (US $) per steer obtained due to the use of growth promoting implants (Implant cost + Labor = US $3.2 per animal. Sale price US $1.7 per kg of BW (Auction Osorno FEGOSA; HGP = hormonal growth promoting)).

**Table 1 animals-09-00587-t001:** Nutritional characteristics of the pasture grazed in the study ^1^.

Item	March	September	Octuber	November
DM content, %	16.38	14.48	15.41	16.95
Crude Protein (CP), %	23.04	28.29	26.08	23.22
Neutral Fiber Detergent (NDF), %	46.06	38.67	39.22	41.70
Acid Fiber Detergent (ADF), %	22.64	19.76	21.20	21.45
ME, Mcal/kg DM	2.65	2.75	2.77	2.76
NEm, Mcal/kg DM	1.73	1.82	1.84	1.83
NEg, Mcal/kg DM	1.12	1.19	1.21	1.20

^1^ Permanent pasture fertilized, Osorno [13]. ME = metabolizable energy; NEm = Net energy of maintenance and NEg = net energy of gain.

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
