# Peer review of "Effect of High Potency Growth Implants on Average Daily Gain of Grass-Fattened Steers"

_animals, 2019, doi:10.3390/ani9090587_

Round 1
Reviewer 1 Report
The manuscript documents novel research that examines the use of growth implants in grass-fattened cattle. There are few reports of how growth implants may enhance growth rates in grass-fattened steers. The manuscript is well-written and only a few minor editorial comments are needed. One addition to the manuscript would be the exact growth implants that were used for this study. There are many Revalor implants that can be used in steers so the exact one and hormone levels are needed for the reader to understand which type of implant was used in this experiment. The hormone levels should also be included for the Synovex-Plus implant as well.
Specific comments:
25 Label the treatment with each ADG level for clarity.
40 Change back eye area to ribeye area
41 Change carcass is obtained to carcass weight obtained.
89-90 The exact implant used in this experiment should be provided along with hormonal content.
126 What is PC? should it be cruse protein?
156 Change y to and.
155 and 157 The check symbol for plus/minus.
192 deleted s on variable.
206 The hormonal levels are not correct for the Revalor. There is no zeranol in revalor. Revalor-G is 40 mg TBA and 8 mg estradiol. Revalor-S is 120 mg TBA and 24 mg estradiol. Revalor H is 140 mg TBA and 14 mg estradiol. Authors need to define which implant was used and correctly list hormonal levels.
227 Please expand on the 16% overpriced comment. I do not understand what you mean here.
Author Response
We submitted a file with our answers.

Reviewer 2 Report
This paper reports on a nicely designed trial investigating the effect of HGPs on the liveweight gain of grazing steers. The results are quite clear and the discussion extensive.
I have a number of issues which need to be addressed and some questions as follows.
In the methods the active ingredients of the HGPs must be stated
In Table 1, MS needs to be changes to DM
In Figure 3, where did the 'd' subscript come from?
Given the evidence that HGPs can influence dressing out%, thus carcass weight and subsequently carcass grade and meat quality, why were these cattle not slaughtered at the end of the trial? If they were grown on after the end of the HGP period, the opportunity for compensatory growth in the Control group of steers must receive comment.
It is always dangerous to incorporate financial data in a formal paper as these change with time and become outdated. Such an analysis is fine for a Farmers Day presentation but not a formal paper.
Although I haven't had the time I am sure I could locate other reports of the use of HGPs in grazing cattle.
I question the wisdom of undertaking this work given the increasing number of markets for beef that are not accepting beef from HGP treated cattle.
Your paper needs another revision to decrease its length and eliminate surplus words and improve the English grammar.
Thank you for the opportunity to review your paper.
Author Response
We submitted a file with our answers
